# Five-Hour Detection of Intestinal Colonization with Extended-Spectrum-$\beta$-Lactamase-Producing *Enterobacteriaceae* Using the $\beta$-Lacta Phenotypic Test: the BLESSED Study

Salah Gallah,[a] Maximilien Scherer,[b] Thierry Collin,[a] Camille Gomart,[a] Nicolas Veziris,[a,c] Yahia Benzerara,[a] (ID) Marc Garnier[b]

[a]Département de Bactériologie, Hôpital Saint-Antoine, AP-HP, Sorbonne Université, Paris, France
[b]Sorbonne Université, APHP.6, GRC29, DMU DREAM, Département d'Anesthésie-Réanimation et Médecine Périopératoire—Site Tenon, Paris, France
[c]Centre d'Immunologie et des Maladies Infectieuses, INSERM, U1135, Sorbonne Université, Paris, France

Salah Gallah and Maximilien Scherer contributed equally to this work. Author order was determined in order of seniority.

**ABSTRACT** Extended-spectrum-$\beta$-lactamase (ESBL)-producing *Enterobacteriaceae* (ESBL-PE) intestinal colonization is of particular concern as it negatively impacts morbidity and is the main source of external cross-contamination in hospitalized patients. Contact isolation strategies may be caught out due to the turnaround time needed by laboratories to report intestinal colonization, during which patients may be inappropriately isolated or not isolated. Here, we developed a protocol combining enrichment by a rapid selective subculture of rectal swab medium and realization of a $\beta$-Lacta test on the obtained bacterial pellet (named the BLESSED protocol). The performances of this protocol were validated *in vitro* on 12 ESBL-PE strains spiked into calibrated sample suspensions and confirmed in clinical settings using 155 rectal swabs, of which 23 (reference method) and 31 (postenrichment broth culture) came from ESBL-PE carriers. *In vitro*, the protocol detected, with 100% sensitivity, the presence of the 12 ESBL-PE strains from $10^4$ CFU/mL. In the clinical validation cohort, 22 out of the 23 (reference method) and 28 out of the 31 (postenrichment broth culture) ESBL-PE-positive rectal samples were accurately detected. The diagnostic performances for ESBL-PE detection, considering all ESBL-PE carriers, were 90% sensitivity, 98% specificity, an 87% positive predictive value, and a 98% negative predictive value. Our protocol is a rapid and low-cost method that can detect intestinal colonization with ESBL-PE in less than 5 h more accurately than the reference method, opening the field for further studies assessing a rapid and targeted isolation strategy applied only to patients with a positive BLESSED protocol result.

**IMPORTANCE** To both improve the efficiency of contact isolation among ESBL-PE carriers and avoid the unnecessary isolation of noncolonized patients, we should reduce the turnaround time of ESBL screening in laboratories and improve the sensitivity of diagnostic methods. The development of rapid and low-cost methods that satisfy these two goals is a promising approach. In this study, we developed such a technique and report its good diagnostic performance, opening the door for further studies assessing a rapid and targeted isolation strategy applied in a few hours only for patients truly colonized with ESBL-producing bacteria.

**KEYWORDS** extended-spectrum $\beta$-lactamase, intestinal colonization, phenotypic test, rapid detection, diagnostic performance

Address correspondence to Marc Garnier, marc.garnier@aphp.fr.

The authors declare no conflict of interest.

The extensive use of $\beta$-lactam antimicrobials during the past 2 decades has led to the selection of extended-spectrum-$\beta$-lactamase (ESBL)-producing *Enterobacteriaceae* (ESBL-PE). The rate of ESBL-PE intestinal colonization in intensive care units (ICUs) in France can reach 25% (1) and is linked directly to an increase in the hospital length of stay and indirectly

to hospital mortality and the cost of treatment of ESBL-PE-related infections (2). Moreover, this colonization often compels the use of broad-spectrum antibiotics, such as carbapenems, favoring the emergence of carbapenemase-producing Gram-negative bacilli (GNB) (3). Therefore, ESBL-PE colonization constitutes a major public health concern, and its screening is a priority (4, 5). Current conventional screening strategies are based mainly on culture on selective agar plates, followed by the identification and confirmation of the antibiotic resistance phenotype of the bacterial strains. Some technical advances, such as automated specimen inoculation and plate streaking (6), plate reading (7), and bacterial identification using matrix-assisted laser desorption ionization–time of flight (MALDI-TOF) mass spectrometry, allow for saving some time in the entire procedure. However, in any case, the main limitation of such strategies remains the time required to obtain a definite result, which is approximately 48 to 72 h (8). Several strategies to prevent cross-contamination from patients colonized with ESBL-PE pending the definite microbiological results are used in ICU clinical practice: (i) patients are not preventively systematically isolated, with only the patients who are truly colonized being isolated after the results of screening; (ii) all patients are preventively systematically isolated and then secondarily nonisolated if the results of screening are negative; or (iii) only some patients are preventively isolated based on the presence of risk factors for ESBL-PE colonization, sometimes compiled in risk scores (9). However, all three strategies are questionable and may lead to the unnecessary isolation of noncolonized patients or, on the contrary, the nonisolation of colonized patients, which may negatively affect the quality of care (10). Thus, the rapid detection of ESBL-PE colonization can improve the isolation strategy, reducing both the inappropriate isolation and nonisolation of patients (10, 11). In addition, in the ICU setting, the rapid determination of the ESBL-PE colonization status of a patient suffering from severe infection can limit the inappropriate use of broad-spectrum antibiotics during the empirical phase.

Many diagnostic tests using phenotypic, immunochromatographic, and molecular methods have been developed to detect the presence of ESBL-PE, and several of these tests have shown high sensitivity and specificity (12–15). Molecular biology can be performed directly on rectal samples with good sensitivity and specificity, providing results in only a few hours (16). However, this method is expensive, requires dedicated equipment, and, thus, may not be affordable for all laboratories. In contrast, phenotypic tests are simple to use and less expensive. The $\beta$-Lacta test (BLT) (Bio-Rad Laboratories, Marnes-La-Coquette, France) is a phenotypic test based on the cleavage of a chromogenic cephalosporin, turning the substrate from yellow to red in the presence of GNB strains producing a $\beta$-lactamase that is able to hydrolyze third-generation cephalosporins. Thus, the test turns positive mainly in the presence of ESBL-PE; however, the test also reacts with some carbapenemase-producing and AmpC-hyperproducing GNB. The BLT has shown high diagnostic performance for the rapid identification of ESBL-PE when performed on isolated bacterial colonies (17, 18) or directly on bacterial pellets from urine or bronchial aspirate samples (12, 14). However, it has never been assessed for the rapid detection of ESBL-PE intestinal colonization.

In the present study, we developed a protocol to allow the rapid detection of intestinal colonization with ESBL-PE using the BLT performed directly on a short selective subculture of a rectal swab.

## RESULTS

***In vitro* performance of the BLESSED protocol on mock-calibrated rectal swabs. (i) Performance of the selective enrichment broth.** The ESBL-PE inocula spiked into the eSwab medium were enriched 84-fold (25th to 75th percentiles, 23- to 318-fold) by the 4-h subculture in selective medium. The enrichment factor was independent of the preenrichment bacterial inoculum (75-fold [42- to 213-fold] versus 73-fold [17- to 155-fold] for the $10^3$- and $10^4$-CFU/mL inocula, respectively; $P = 0.89$). The enrichment factor values were significantly higher for *Escherichia coli* and *Klebsiella pneumoniae* strains than for other *Enterobacteriaceae* species (323-fold [190- to 587-fold] and 57-fold [45- to 72-fold] versus 5-fold [2- to 19-fold], respectively; $P < 0.001$). No difference was found in the enrichment factors depending on the type of

**TABLE 1** *In vitro* evaluation of the BLESSED protocol for the detection of 16 ESBL-PE strains spiked at $10^3$ and $10^4$ CFU/mL into calibrated swab liquid sample suspensions

| GNB strain | ESBL enzyme produced | CFU/mL | Enrichment factor (fold) | BLT result at: | | | |
|---|---|---|---|---|---|---|---|
| | | | | 15 min | 30 min | 45 min | 60 min |
| *Klebsiella pneumoniae* | CTX-M-22 | $10^3$ | 350 | − | − | + | + |
| | | $10^4$ | 540 | + | + | + | + |
| *Klebsiella pneumoniae* | CTX-M-3 | $10^3$ | 160 | − | − | − | − |
| | | $10^4$ | 740 | + | + | + | + |
| *Klebsiella pneumoniae* | CTX-M-15 | $10^3$ | 310 | + | + | + | + |
| | | $10^4$ | 330 | + | + | + | + |
| *Klebsiella pneumoniae* | SHV-12 | $10^3$ | 500 | − | − | − | − |
| | | $10^4$ | 930 | − | + | + | + |
| *Klebsiella pneumoniae* | TEM-3 | $10^3$ | 250 | − | − | − | − |
| | | $10^4$ | 96 | − | + | + | + |
| *Klebsiella pneumoniae* | TEM-21 | $10^3$ | 200 | − | − | − | − |
| | | $10^4$ | 20 | + | + | + | + |
| *Escherichia coli* | CTX-M-27 | $10^3$ | 43 | − | − | − | − |
| | | $10^4$ | 7 | + | + | + | + |
| *Escherichia coli* | CTX-M-1 | $10^3$ | 50 | + | + | + | + |
| | | $10^4$ | 64 | − | − | + | + |
| *Escherichia coli* | SHV-2 | $10^3$ | 75 | − | − | − | − |
| | | $10^4$ | 93 | − | − | − | + |
| *Citrobacter freundii* | CTX-M-3 | $10^3$ | 36 | − | − | − | − |
| | | $10^4$ | 1.3 | − | − | + | + |
| *Citrobacter freundii* | TEM-3 | $10^3$ | 5 | − | − | − | − |
| | | $10^4$ | 24 | − | − | − | + |
| *Enterobacter cloacae* | CTX-M-15 | $10^3$ | 2 | − | − | − | − |
| | | $10^4$ | 5 | − | + | + | + |

ESBL enzyme (57-fold [14- to 328-fold] versus 95-fold [37- to 238-fold] for CTX-M and non-CTX-M, respectively; *P* = 0.58) (Table 1).

**(ii) Performance of the BLT.** The BLT, read after 60 min, was positive for only 3 out of the 12 ESBL-PE strains spiked as a $10^3$-CFU/mL inoculum (25% sensitivity); however, it was positive for all 12 strains spiked as a $10^4$-CFU/mL inoculum (100% sensitivity) (Table 1). Therefore, the detection threshold of the BLESSED protocol was probably between $10^3$ and $10^4$ CFU/mL.

**Clinical validation of the performance of the BLESSED protocol. (i) Population.** A total of 155 rectal swabs from 155 different patients in the ICU were collected. No difference was observed in demographic characteristics between the patients colonized (*n* = 31; incidence, 20%) and those not colonized with ESBL-PE (Table 2). Patients colonized with ESBL-PE presented more clinical risk factors (3 risk factors [25th to 75th percentiles, 1 to 3.5] versus 1 [0 to 2]; *P* = 0.01). In particular, they were more frequently hospitalized in the previous 12 months or colonized with ESBL-PE in the last 6 months, or they required chronic renal replacement (Table 2).

**(ii) Microbiological data.** In total, 106 out of the 155 rectal swab samples showed a negative culture on selective agar plates. Among the 49 positive samples, cultures yielded 25 cephalosporinase-hyperproducing (H-CASE) GNB strains, 1 wild-type *Stenotrophomonas maltophilia* strain, and 23 ESBL-PE strains (*E. coli*, *n* = 12; *K. pneumoniae*, *n* = 6; *Enterobacter cloacae*, *n* = 5) (Table 3). In addition, 8 of the 106 negative samples after culture on selective

**TABLE 2** Patient characteristics[a]

| | Value for group | | | |
| --- | --- | --- | --- | --- |
| Parameter | All patients (n = 155) | Patients without ESBL-PE (n = 124) | Patients with ESBL-PE (n = 31) | P value |
| Demographic characteristics | | | | |
| Median patient age (yrs) (25th–75th percentiles) | 64 (54–74) | 65 (54–74) | 61 (55–76) | 0.90 |
| No. of patients of sex (M/F) | 105/50 | 82/42 | 23/8 | 0.39 |
| Median simplified acute physiology score II (25th–75th percentiles) | 32 (21–45) | 31 (19–46) | 36 (25–44) | 0.22 |
| No. (%) of patients with type of ICU admission | | | | |
| Medical | 79 (50) | 63 (50) | 16 (52) | 0.81 |
| Surgical, urgent | 35 (22) | 27 (22) | 8 (26) | |
| Surgical, scheduled | 41 (26) | 34 (28) | 7 (22) | |
| No. (%) of patients with cause(s) of ICU admission | | | | |
| Sepsis, septic shock | 49 (31) | 39 (31) | 10 (32) | 0.06 |
| Acute respiratory failure | 13 (8) | 7 (9) | 4 (12) | |
| Postoperative care | 49 (32) | 40 (32) | 9 (29) | |
| Hemorrhagic shock | 33 (21) | 26 (21) | 7 (23) | |
| Coma, epilepsy, neurological failure | 6 (4) | 36 (5) | 0 (0) | |
| Cardiogenic shock/cardiorespiratory arrest | 5 (3) | 4 (3) | 1 (3) | |
| No. (%) of patients with ESBL-PE colonization risk factor | | | | |
| Hospitalization for ≥48 h in the previous 12 mo | 91 (58) | 65 (52) | 26 (84) | 0.01 |
| Antibiotic therapy in the previous 3 mo | 58 (37) | 43 (35) | 15 (48) | 0.16 |
| Hospital stay of ≥5 days before ICU admission | 55 (35) | 40 (32) | 15 (48) | 0.09 |
| Immunosuppression | 27 (17) | 23 (18) | 4 (13) | 0.46 |
| Colonization with ESBL-PE in the previous 6 mo | 21 (13) | 10 (8) | 11 (35) | <0.001 |
| Life in an institution or nursing home | 8 (5) | 6 (5) | 2 (6) | 0.72 |
| Travel in a zone where ESBL-PE are endemic in the previous 12 mo | 8 (5) | 6 (5) | 2 (6) | 0.72 |
| Chronic renal replacement | 4 (3) | 0 (0) | 4 (13) | <0.001 |
| Recurrent urinary tract infections | 1 (1) | 1 (1) | 0 (0) | 0.61 |
| Total no. of risk factors per patient (25th–75th percentiles) | 2 (0–3) | 1 (0–2) | 3 (1–3.5) | 0.01 |
| Median length of stay (days) (25th–75th percentiles) in: | | | | |
| ICU | 3 (2–5.5) | 3 (2–5) | 3 (2–6) | 0.80 |
| Hospital | 12 (7–22) | 12 (7–20) | 11 (4–18) | 0.37 |

[a]M, male; F, female.

agar plates actually contained ESBL-PE strains, as identified by cultures of the postenrichment selective broth (E. coli, n = 3; K. pneumoniae, n = 3; E. cloacae, n = 1; Klebsiella oxytoca, n = 1) (Table 4). Thus, the prevalences of ESBL-PE intestinal colonization were 15% (23/155) considering the reference method and 20% (31/155) considering ESBL-PE detected using a postenrichment broth culture.

Among the 31 ESBL-PE strains, 28 produced a CTX-M-type enzyme (19 CTX-M-15, 5 CTX-M-27, 3 CTX-M-3, and 1 CTX-M-1), whereas 3 isolates produced a TEM-type enzyme.

**(iii) Performance of the BLESSED protocol.** Compared with the reference method, the BLESSED protocol was positive for 22 out of the 23 ESBL-PE, 3 out of the 25 H-CASE GNB, and 6 out of the 106 negative cultures (Table 3), providing 96% sensitivity, 93% specificity, a 71% positive predictive value, and a 99% negative predictive value (Table 5).

Compared with the culture of the postenrichment selective broth, the BLESSED protocol was positive for 28 out of the 31 ESBL-PE, 3 out of the 25 H-CASE, and none of the negative cultures (Table 4), providing 90% sensitivity, 98% specificity, an 87% positive predictive value, and a 98% negative predictive value (Table 5). In contrast, direct culture of the rectal swab medium (reference method) showed 74% sensitivity, 100% specificity, a 100% positive predictive value, and a 92% negative predictive value.

**Comparison of the performances of the clinical risk score and the BLESSED protocol in predicting ESBL-PE intestinal colonization.** The performance of the clinical risk score for predicting ESBL-PE colonization was poor, with an area under the receiver operating characteristic (ROC) curve (AUROC) of 0.70 (95% confidence interval [CI], 0.57 to 0.83) (Fig. 1). A clinical

**TABLE 3** Clinical evaluation of the BLESSED protocol using 155 rectal samples for the detection of ESBL-PE compared with culture of rectal swab medium on ESBL chromID agar plates (reference method)[a]

| Isolated bacterium or BLESSED protocol result | No. of samples with culture result | | | |
|---|---|---|---|---|
| | Negative | Positive for: | | |
| | | ESBL-PE | H-CASE GNB | Other |
| **Isolated bacterium** | | | | |
| None | 106 | | | |
| *Escherichia coli* | | 12 | 3 | 0 |
| *Klebsiella pneumoniae* | | 6 | 0 | 0 |
| *Enterobacter cloacae* | | 5 | 11 | 0 |
| *Klebsiella oxytoca* | | 0 | 0 | 0 |
| *Citrobacter freundii* | | 0 | 1 | 0 |
| *Klebsiella aerogenes* | | 0 | 1 | 0 |
| *Pseudomonas aeruginosa* | | 0 | 8 | 0 |
| *Acinetobacter baumannii* | | 0 | 1 | 0 |
| *Stenotrophomonas maltophilia* | | 0 | 0 | 1 |
| Total | 106 | 23 | 25 | 1 |
| **BLESSED protocol result** | | | | |
| Positive | 6 | 22 | 3 | 0 |
| Negative | 100 | 1 | 22 | 1 |

[a]ESBL-PE, extended-spectrum-$\beta$-lactamase-producing *Enterobacteriaceae*; H-CASE GNB, cephalosporinase-hyperproducing Gram-negative bacilli.

score of ≥3 had 61% sensitivity, 76% specificity, and a positive likelihood ratio of 2.5 for predicting ESBL-PE intestinal colonization. The AUROC of the BLESSED protocol was significantly better (0.97 [0.95 to 1.00]; $P < 0.001$ for comparison) (Fig. 1).

In the univariable analysis, both a clinical risk score of ≥3 (odds ratio [OR], 1.65 [95% CI, 1.21 to 2.30]) and a positive BLESSED result (OR, 363 [69.9 to 3,107]) were associated with ESBL-PE colonization; however, only a positive BLESSED result was independently associated with ESBL-PE colonization in the multivariable analysis (OR, 320 [52.9 to 6,367]; $P < 0.001$) (Table 6).

**TABLE 4** Clinical evaluation of the BLESSED protocol using 155 rectal samples for the detection of ESBL-PE compared with culture of postenrichment broth on ESBL chromID agar plates[a]

| Isolated bacterium or BLESSED protocol result | No. of samples with culture result | | | |
|---|---|---|---|---|
| | Negative | Positive for: | | |
| | | ESBL-PE | H-CASE GNB | Other |
| **Isolated bacterium** | | | | |
| None | 98 | | | |
| *Escherichia coli* | | 16 | 3 | 0 |
| *Klebsiella pneumoniae* | | 8 | 0 | 0 |
| *Enterobacter cloacae* | | 6 | 11 | 0 |
| *Klebsiella oxytoca* | | 1 | 0 | 0 |
| *Citrobacter freundii* | | 0 | 1 | 0 |
| *Klebsiella aerogenes* | | 0 | 1 | 0 |
| *Pseudomonas aeruginosa* | | 0 | 8 | 0 |
| *Acinetobacter baumannii* | | 0 | 1 | 0 |
| *Stenotrophomonas maltophilia* | | 0 | 0 | 1 |
| Total | 98 | 31 | 25 | 1 |
| **BLESSED protocol result** | | | | |
| Positive | 0 | 28 | 3 | 0 |
| Negative | 98 | 3 | 22 | 1 |

[a]ESBL-PE, extended-spectrum-$\beta$-lactamase-producing *Enterobacteriaceae*; H-CASE GNB, cephalosporinase-hyperproducing Gram-negative bacilli.

**TABLE 5** Performance characteristics of the BLESSED protocol for the detection of ESBL-PE in clinical settings

| Diagnostic performance characteristic | Median value (95% CI) compared with: | |
| --- | --- | --- |
| | Rectal swab culture | Postenrichment broth culture |
| Sensitivity (%) | 96 (78–100) | 90 (74–98) |
| Specificity (%) | 93 (87–97) | 98 (93–100) |
| Positive predictive value (%) | 71 (57–82) | 87 (68–95) |
| Negative predictive value (%) | 99 (95–100) | 98 (95–99) |
| Positive likelihood ratio | 14 (7.4–27) | 37.3 (12.1–114.8) |
| Negative likelihood ratio | 0.05 (0.01–0.32) | 0.10 (0.03–0.29) |

## DISCUSSION

This study reports the first utilization of the BLT on digestive tract samples, with a simple and reproducible protocol showing good diagnostic performance in identifying ESBL-PE intestinal colonization in less than 5 h. Our protocol even detected six out of eight ESBL-PE diagnosed by postenrichment broth culture that were not detected by conventional culture of rectal swab medium, probably due to a low digestive inoculum. This finding is concordant with previous results reporting the lack of sensitivity of direct cultures of rectal swabs in identifying all ESBL-PE carriers (19). The only three false-negative results observed in our study with the BLESSED protocol corresponded to very low inocula of $<10^2$ CFU/mL. Thus, considering *in vitro* and clinical evaluations, the BLESSED protocol had a sensitivity of >95% in detecting ESBL-PE in an inoculum of $\geq 10^4$ CFU/mL, which is better than the reference method.

The detection of intestinal colonization with ESBL-PE is helpful in controlling their spread, as fecal carriage is the major source of the exogenous transmission of ESBL-PE during hospital stays (20, 21). Despite doubts regarding their efficacy (22), many centers have applied contact precautions, in addition to standard precautions, to prevent cross-transmission from patients colonized with ESBL-PE (23, 24), according to some guidelines (25, 26). However, one of the weaknesses of this isolation strategy is the laboratory turnaround time of at least 2 days, leading to delays in reporting ESBL-PE-positive samples and, consequently, implementing contact isolation. In a large cluster-randomized study, this delay represented approximately one-third of ESBL-PE patient days during which contact precautions were not applied (22). To avoid this flaw, many centers have also applied contact precautions in newly admitted patients while waiting for the results of their rectal swab screening. If this strategy may prevent cross-transmission before a patient's colonization status is determined, it presents several disadvantages. First, it leads to many unnecessary isolations, which are sources of additional tasks for health care professionals and supplementary costs. Indeed, each day of preventive contact isolation has been estimated to cost €155 ($160) per patient (27). In addition,

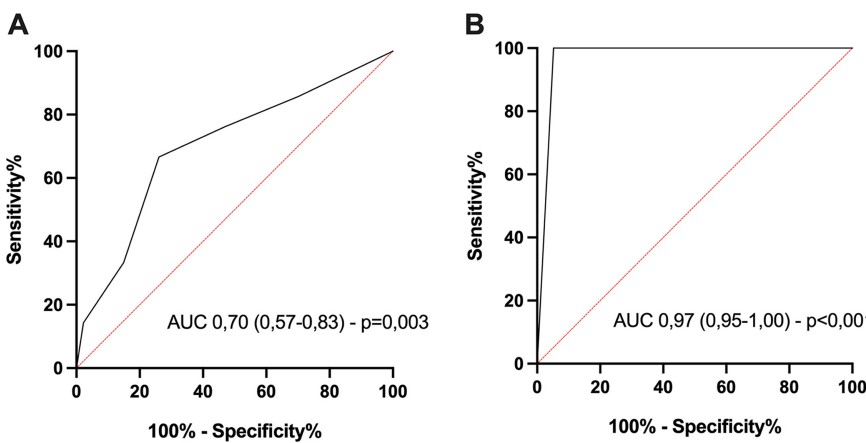

**FIG 1** Receiver operating characteristic (ROC) curves of the clinical risk score (A) and the BLT (B) for the detection of ESBL-PE intestinal colonization. AUC, area under the concentration-time curve.

**TABLE 6** Factors associated with ESBL-PE colonization

| Factor | Univariable analysis | | Multivariable analysis | |
|---|---|---|---|---|
| | OR (95% CI) | P value | OR (95% CI) | P value |
| Clinical risk score of ≥3 | 1.65 (1.21–2.30) | 0.002 | 1.14 (0.65–1.98) | 0.65 |
| Positive BLESSED result | 363 (69.9–3,107) | <0.001 | 320 (52.9–6,367) | <0.001 |

patients under contact isolation receive fewer visits from health caregivers, who spend approximately 20% less time with such patients (28), leading to a significant increase in adverse events such as errors in applying prescriptions, hypoglycemia, or ventilator-associated pneumonia (29, 30). Finally, contact isolation also negatively affects the patients' mental well-being and behavior, generating anxiety and anger (30). Consequently, the best strategy may be to avoid unnecessary contact isolation and nonisolation of colonized patients. For this purpose, some studies have proposed targeted isolation and/or screening based on the presence of several clinical risk factors (9, 31). However, in this study, we show that such an approach guided by risk scores is highly flawed, with poor sensitivity and specificity and a lack of an association with ESBL-PE colonization in the multivariable analysis. In contrast, the BLESSED protocol, based on a short selective subculture of a rectal swab coupled with a phenotypic test, may allow us to reach this double goal thanks to its high negative predictive value, which allows us to avoid the unnecessary isolation of patients with a negative BLESSED result, and its acceptable positive predictive value, which leads to the appropriate isolation of almost all patients colonized with ESBL-PE and the excessive temporary isolation of a few patients colonized with third-generation-cephalosporin-resistant GNB, mainly due to cephalosporinase hyperproduction. When considering postenrichment broth culture, the sensitivity of the BLESSED protocol in clinical settings appeared better than that of direct culture of rectal swabs, with the latter being similar to that reported previously by Jazmati et al. (19). Furthermore, the performance of our protocol is close to the excellent diagnostic performance reported by Blanc et al. using other phenotypic tests to screen carriers of ESBL-PE (32). In that study, tests were performed on fresh colonies obtained after culture overnight on agar plates, leading to a turnaround time of 36 to 38 h to report an ESBL-PE-positive sample (32), which is slightly shorter than the 48 h to 72 h usually required for the reference method. While clinically relevant, our protocol allowed the turnaround time to be reduced to less than 5 h. This may open the door to a strategy in which the contact isolation of patients would be applied only in cases of a positive BLESSED protocol result, thus allowing rapid, targeted isolation. Finally, the last advantage of this protocol is its relatively low cost. Indeed, phenotypic tests, such as the BLT, cost only a few euros, while selective enrichment broth is no more expensive. Consequently, such a protocol is much less expensive than molecular techniques and requires less material, which allows it to be used anywhere regardless of the level of equipment in the laboratory. Moreover, current molecular techniques have not shown better diagnostic performances than those reported for the BLESSED protocol in this study (16, 33).

Our study has several limitations. First, our protocol requires a 4-h incubation phase, which extends the total duration of the procedure. However, this selective enrichment step is crucial as it greatly enhances the performance of the protocol by amplifying low inocula and eliminating contamination. Indeed, the BLT used directly on bacterial pellets from rectal swab transport medium showed an insufficient diagnostic performance, and only ESBL-PE inocula of >10⁵ CFU/mL were accurately detected. In addition, contamination of the liquid medium of the rectal swab with cellular and bacterial debris, as well as digestive anaerobic flora, led to numerous false-positive results. Due to both the selective and nutritive functions of our broth, ESBL-PE inocula were specifically amplified as the diagnostic performance of the final protocol. Second, our protocol requires several manipulations. Although the number is quite limited, it could be hypothesized that the automation of the procedure would increase its diffusion. However, this must be balanced with the fact that the reference screening strategy is also not fully automated. Indeed, the plating of rectal swab medium onto selective agar plates, the reading of the culture results

of these plates, and, in cases of positivity, the subculturing of isolated strains to determine their antibiotic susceptibility remain mostly manual. Thus, the BLESSED protocol could probably already be implemented in most laboratories, without a significant increase in the technical handling time but allowing the detection of ESBL-PE intestinal colonization from patient sampling to final results in less than a working day for a laboratory technician.

In conclusion, the BLESSED protocol is a rapid, easy-to-perform, and low-cost method that allows the accurate detection of intestinal colonization with ESBL-PE in less than 5 h after rectal sampling, without any requirement for specific equipment. These results open the door for further studies evaluating the impact of the rapid phenotypic diagnosis of ESBL-PE intestinal colonization in terms of the targeted contact isolation of patients and the prevention of cross-transmission.

## MATERIALS AND METHODS

**Technical optimization procedure.** We first assessed the performance of the BLT directly on bacterial pellets obtained from the centrifugation of 1 mL of rectal swab medium (see Text S1 in the supplemental material for additional details). The pellets were completely homogenized in 1 drop (approximately 50 $\mu$L) of each reagent from the BLT and incubated at 37°C. This "direct" procedure showed poor diagnostic performance in detecting ESBL-PE (sensitivity of 60% [95% confidence interval {CI}, 26% to 88%], specificity of 58% [42% to 72%], positive predictive value of 24% [15% to 37%], and negative predictive value of 87% [75% to 94%]). These results can be explained by the low sensitivity in detecting ESBL activity in samples diluted in the swab transport medium and the low specificity due to false-positive results related to other bacteria that can turn the BLT positive, such as some bacteria of the anaerobic flora. We then developed a rapid, selective subculture technique to increase both the sensitivity and specificity of the procedure.

Rapid selective subculture was first validated on rectal samples spiked with calibrated concentrations of ESBL-PE strains. Mock-calibrated rectal swab medium suspensions were prepared by mixing 1 mL of eSwab liquid medium with 100 mg of a pool of stool specimens free of any ESBL- and carbapenemase-producing GNB. The mock suspensions were then spiked with two concentrations of ESBL-PE strains collected from different French hospitals (final rectal swab medium concentrations of $10^3$ and $10^4$ CFU/mL; 12 different genes for 9 different variants). Five hundred microliters of the spiked mock rectal swab suspension was added to 4.5 mL of $\beta$-lactamase-selective enrichment broth (brain heart infusion broth containing an antibiotic mix [patent number IDDN.FR.001.200006.000.S.P.2021.000.31230] (38)) and incubated for 2, 3, or 4 h at 37°C under constant agitation. Next, 2 mL of the selective enrichment broth was centrifuged for 5 min at 10,000 $\times$ $g$. The supernatant was discarded, the bacterial pellet was mixed with 1 drop of each BLT-specific reagent, and this final suspension was incubated at 37°C. The results were read after 15, 30, 45, and 60 min by two independent observers. The result was interpreted as positive in the case of a colorimetric shift from yellow to orange or red. All experiments were performed in duplicate.

The enrichment performance of the selective broth was documented by comparing bacterial counts obtained after 24 h of culture on $\beta$-lactamase-selective agar plates (chromID BLSE; bioMérieux, Marcy-L'Étoile, France) of the selective enrichment broth before and after the 4-h subculture.

The final protocol that combined the experimental conditions providing the highest sensitivity for the detection of ESBL-PE was named the "$\beta$-Lacta test on enriched subcultures of rectal swabs for ESBL-PE detection (BLESSED) protocol." The BLESSED protocol is based on a 4-h enrichment in selective broth, followed by the performance of the BLT on the centrifuged bacterial pellet and reading of the results after 60 min (see the supplemental material for detailed results and figures concerning the determination of the best experimental conditions).

**Evaluation of the performance of the BLESSED protocol in clinical settings. (i) Collection.** Unused residual rectal eSwab media for laboratory diagnosis performed to screen for intestinal colonization with multidrug-resistant GNB in 155 patients hospitalized in the ICUs of two university hospitals in Paris (AP-HP Tenon and Saint-Antoine hospitals) were prospectively collected from June 2019 to October 2020 and frozen at −20°C until further use. This collection was approved by the Nord-Ouest III Ethics Committee (approval ID RCB 2017-A02339-44).

**(ii) Bacterial culture, identification, and characterization.** Fifty microliters of swab transport medium was spread onto $\beta$-lactamase-selective agar plates (chromID ESBL) and incubated for 24 h at 37°C. A rectal swab culture was considered positive when the culture yielded at least one colony ($\geq$20 CFU/mL) of ESBL-PE. Bacterial identification was performed using MALDI-TOF mass spectrometry (Biotyper; Bruker Daltonics, Billerica, MA, USA). The results of $\beta$-lactam susceptibility testing using Kirby-Bauer disk diffusion were interpreted after 24 h of subculture of colonies growing on the chromID ESBL plates, according to the recommendations of the European Committee on Antimicrobial Susceptibility Testing (EUCAST) and the Antibiotic Committee of the French Society of Microbiology (CA-SFM) (34). The detection and typing of ESBL genes were performed by amplification and sequencing, as previously described (35). ESBL sequences were identified using the Beta-Lactamase Database described previously by Naas et al. (36).

**(iii) Realization of the BLT.** The BLT was performed on each rectal swab sample after 4 h of enrichment, and its results were read after 60 min, according to the BLESSED protocol. The diagnostic performances of the BLESSED protocol were then compared with those of direct culture of the rectal eSwab medium on chromID ESBL plates (reference method) and the postenrichment selective broth.

**Clinical risk score for ESBL-PE colonization.** To assess the relevance of an isolation strategy based on clinical risk factors for ESBL-PE colonization, we compiled nine risk factors from previous studies (9, 10, 37) into

a clinical risk score (Table 1; see also Text S1 in the supplemental material for additional details). The diagnostic performance of this clinical score was evaluated against the reference method.

**Statistical analysis.** Quantitative data are expressed as median values (25 to 75th percentiles), and qualitative data are expressed as numbers (percentages). Performance characteristics are given as values (95% CIs). Distributions were compared between the ESBL and non-ESBL groups using the Mann-Whitney-Wilcoxon test and the chi-square or Fisher exact test for continuous and qualitative variables, respectively. To identify the factors associated with ESBL-PE intestinal colonization, we included all clinically relevant covariates in a multivariable linear regression model. All tests were two tailed, and $P$ values of $<0.05$ were considered significant. Statistical analysis was performed using GraphPad Prism 8.4.3 (GraphPad Software, San Diego, CA, USA).

**Data availability.** Raw data are available from the corresponding author upon reasonable request.

## SUPPLEMENTAL MATERIAL

Supplemental material is available online only.

**SUPPLEMENTAL FILE 1**, PDF file, 0.2 MB.

## ACKNOWLEDGMENTS

S.G. conceived the study, supervised technical developments and evaluation/validation experiments, and analyzed the data. M.S. performed the evaluation/validation experiments, analyzed the data, and wrote the first draft of the manuscript. T.C. and C.G. performed the technical developments and contributed to evaluation experiments. N.V. helped in analyzing the data and reviewed the manuscript. Y.B. helped in conceiving the study, analyzed the data, and supervised the validation experiments. M.G. helped in conceiving the study, analyzed the data, performed statistical analyses, and wrote the first draft of the manuscript and reviews.

We declare that we do not have any conflict of interest related to this work.

This work has not been funded by any external source. All the experiments were performed using the microbiological laboratory's own funds.

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
