## [Reviewer comments · Microbiology Spectrum]

Microbiology Spectrum

Five-hour detection of intestinal colonization with extended-spectrum beta-lactamase-producing Enterobacteriaceae using the β -LACTA[®] phenotypic test: the “BLESSED study”

Salah Gallah, Maximilien Scherer, Thierry Collin, Camille Gomart, Nicolas Veziris, Yahia Benzerara, and Marc Garnier

Corresponding Author(s): Marc Garnier, Assistance Publique - Hopitaux de Paris

Review Timeline:

Submission Date:	July 29, 2022
Editorial Decision:	October 8, 2022
Revision Received:	November 30, 2022
Accepted:	December 4, 2022

Editor: Tulip Jhaveri

Reviewer(s): Disclosure of reviewer identity is with reference to reviewer comments included in decision letter(s). The following individuals involved in review of your submission have agreed to reveal their identity: Jorge Cervantes (Reviewer #1)

Transaction Report:

DOI: <https://doi.org/10.1128/spectrum.02959-22>

October 8, 2022

Dr. Marc Garnier
APHP - Hopital Tenon
Département d'Anesthésie-Réanimation et Médecine Périopératoire
Paris
France

Re: Spectrum02959-22 (5-hour detection of digestive colonization with extended-spectrum beta-lactamase-producing Enterobacteriaceae using the β -LACTA[®] phenotypic test: the "BLESSED study")

Dear Dr. Marc Garnier:

Thank you for submitting your manuscript to Microbiology Spectrum. Your manuscript requires major revisions before it can be considered for publication. When submitting the revised version of your paper, please provide (1) point-by-point responses to the issues raised by the reviewers as file type "Response to Reviewers," not in your cover letter, and (2) a PDF file that indicates the changes from the original submission (by highlighting or underlining the changes) as file type "Marked Up Manuscript - For Review Only". Please use this link to submit your revised manuscript - we strongly recommend that you submit your paper within the next 60 days or reach out to me. Detailed instructions on submitting your revised paper are below.

Link Not Available

Sincerely,

Tulip Jhaveri

Journals Department
Reviewer comments:

Reviewer #1 (Comments for the Author):

I have reviewed the paper by Gallah et al. Although English is correct, some changes in the writing style may help the manuscript to be more readable. The introduction needs to briefly describe what the beta-LACTA test is. Methods for the blessed protocol need to state the number of patients used. Ideally the reader would like to see a growth curve with the sensitivity of the b-LACTA detection. This hasn't been shown. Otherwise, how do we know the 4 hour shaking in BHI was optimal? I understand the data is somehow "in there" from what is shown in Table 1.

This is important since the study by Blanc et al used 48-72h protocols.

It may be a little misleading to just point out the 100% sensitivity, when the comparisons in Table 4 and 5 show something different.

Reviewer #2 (Comments for the Author):

In this manuscript, Gallah et al. describe the performance of a rapid diagnostic assay to detect ESBL-producing organisms in the gastrointestinal tract (GI) compared to culture on a chromogenic agar with and without enrichment broth. I hope the authors view these comments as being constructive and designed to help strengthen their manuscript.

Major Comments:

Introduction: The first section of this manuscript includes a few lines of introducing the readers to the β -LACTA test (lines 62-65, but there is no information on how the test functions. Is it a molecular test, antigenic test, phenotypic, enzymatic, colorimetric? Please provide a brief description of the β -LACTA test.

Methods: Part of the technical optimization procedure could be moved to the main body of the manuscript as it gives important insight as to how the test functions. The methods section should include a more detailed description of how the β -LACTA test works.

Discussion: Recommend adding discussion for clinical labs on how they can implement this test.

Minor Comments:

Line 43-44: Please describe 'current conventional screening strategies' or recommend moving to lines 56-58.

Reviewer #3 (Comments for the Author):

In the current study, the authors developed a rapid selective enrichment subculture method prior to the commercial β -LACTA{trade mark, serif} test (Bio-Rad Laboratories) to detect the intestinal colonization of Extended-Spectrum Beta-Lactamase-producing Enterobacteriaceae (ESBL-PE) from rectal swab. However, the authors did not carry out a head-to-head experiment to compare the performance of β -LACTA{trade mark, serif} test alone and enrichment subculture with β -LACTA{trade mark, serif} test. Although detection of ESBL-PE is a clinically important issue, it is hard to say that this research has new breakthroughs. In my opinion, I think it's just a small modification for the original protocol.

Major:

1. Native English editing recommended
2. The term "digestive colonization" could be changed to intestinal colonization or intestinal colonization of ESBL-PE.

Minor:

1. Add the full name of "BLT test" "ESBL-PE" in the supplementary methods.

Staff Comments:

Preparing Revision Guidelines

Please return the manuscript within 60 days; if you cannot complete the modification within this time period, please contact me. If

you do not wish to modify the manuscript and prefer to submit it to another journal, please notify me of your decision immediately so that the manuscript may be formally withdrawn from consideration by Microbiology Spectrum.

Spectrum

I have reviewed the paper by Gallah et al.

Although English is correct, some changes in the writing style may help the manuscript to be more readable.

The introduction needs to briefly describe what the beta-LACTA test is.

Methods for the blessed protocol need to state the number of patients used.

Ideally the reader would like to see a growth curve with the sensitivity of the b-LACTA detection. This hasn't been shown. Otherwise, how do we know the 4 hour shaking in BHI was optimal? I understand the data is somehow "in there" from what is shown in Table 1.

This is important since the study by Blanc et al used 48-72h protocols.

It may be a little misleading to just point out the 100% sensitivity, when the comparisons in Table 4 and 5 show something different.

We thank the editor and the reviewers for their comments and the opportunity to revise and improve our manuscript. Please find below our point-by-point responses to these comments.

Reviewer #1

1. I have reviewed the paper by Gallah et al. Although English is correct, some changes in the writing style may help the manuscript to be more readable.

We thank the reviewer for this suggestion. The revised version of the manuscript has been entirely proofread and corrected by a native English speaker.

2. The introduction needs to briefly describe what the beta-LACTA test is.

According to this comment and to comment #2 of reviewer #2, we added two sentences briefly explaining how the β -LACTA test functions, as follows: "In contrast, phenotypic tests are simple to use and less expensive. The β -LACTA™ test (BLT) (Bio-Rad Laboratories, Marnes-La-Coquette, France) is a phenotypic test based on the cleavage of a chromogenic cephalosporin, turning the substrate from yellow to red in the presence of GNB strains producing a beta-lactamase able to hydrolyze 3rd generation cephalosporins. Thus, the test turns positive mainly in the presence of ESBL-PE; however, the test also reacts to some carbapenemase-producing and AmpC-hyperproducing GNB. BLT has shown high diagnostic performance for the rapid identification of ESBL-PE when performed on isolated bacterial colonies (17,18) or directly on bacterial pellets from urine or bronchial aspirate samples (12,14). However, it has never been assessed for the rapid detection of ESBL-PE intestinal colonization." (lines 69-77 of the revised manuscript).

3. Methods for the blessed protocol need to state the number of patients used.

The number of patients (and rectal swabs) on which the BLESSED protocol was validated in clinical settings is reported in the "clinical validation" section of the results, first sentence of the "Population" paragraph: "A total of 155 rectal swabs from 155 different patients in the ICU were collected." (line 165 of the original manuscript, line 184 of the revised version). According to the reviewer's comment, we have also added this information in the method section (section "evaluation of the BLESSED protocol in clinical settings, "collection" paragraph), as follows: "Unused residual media for laboratory diagnosis from rectal eSwab™ performed to screen the intestinal colonization with multidrug-resistant GNB of 155 patients hospitalized in the ICUs of two university hospitals [...]" (lines 122-124 of the revised manuscript).

4. Ideally the reader would like to see a growth curve with the sensitivity of the b-LACTA detection. This hasn't been shown. Otherwise, how do we know the 4-hour shaking in BHI was optimal? I understand the data is somehow "in there" from what is shown in Table 1. This is important since the study by Blanc et al used 48-72h protocols.

We thank the reviewer for this suggestion. We agree that our protocol presents the advantage of providing results faster than Blanc *et al.* protocol. Indeed, Blanc's protocol was performed on fresh colonies obtained after overnight culture on selective agar plates, leading to turnaround time of 36-38h to report ESBL-PE positive sample when our protocol allows identification of ESBL-PE from rectal swabs in 5 hours.

The optimal four-hour incubation time of our BLESSED protocol was determined in preliminary experiments. The reviewer is right, we did not show these preliminary results in the original manuscript. We only mentioned that direct realization of a β -LACTA™ test (BLT) on the pellet obtained from centrifugation of the rectal swab medium (i.e., without any incubation in the selective enrichment broth) led to insufficient performances.

However, it is not so simple to represent in one figure the growth curve of BLT sensitivity as requested but we added figures below that explained our best choice. Indeed, we assessed many experimental conditions including various bacterial inoculums, incubation times of the selective broth, and incubation times in the BLT reagents before reading the results. We have progressed by steps to arrive at the final BLESSED protocol. In particular, when a condition showed poor performance after the first set of experiments, it was dropped to refine the protocol to its final version. Consequently, the number of technical development experiments was not strictly the same for all experimental conditions.

However, we changed several parts of the manuscript and of the supplementary methods file, to better explain why we have retained the four-hour incubation time of the selective broth and the 60-minute reading time for the BLT. These changes include:

- The modification of the sentence dealing with the incubation in the selective enrichment broth in the Methods section, as follows: “Five hundred microliters of spiked mock rectal swab suspension was added [...] and incubated for 2, 3 or 4 h at 37 °C under constant agitation.” (instead of “incubated for 4h at 37°C...”, lines 82-86 of the original manuscript and 100-103 of the revised version).
- The modification of the paragraph summarizing the experimental conditions retained in the BLESSED protocol at the end of the “technical optimization procedure” section of the methods; from: “The final protocol combining the 4-hour enrichment in the selective broth followed by BLT performing on the centrifuged bacterial pellet was named “ β -LACTA test on Enriched Subcultures of rectal Swabs for ESBL-PE Detection (BLESSED) protocol” to “The final protocol that combined the experimental conditions providing the highest sensitivity to detect ESBL-PE was named “ β -LACTA test on Enriched Subcultures of rectal Swabs for ESBL-PE Detection (BLESSED) protocol”. The BLESSED protocol is based on a four-hour enrichment in the selective broth, followed by BLT performing on the centrifuged bacterial pellet, and result reading after 60 min (see the *Supplementary Methods* file online for the detailed results and figures concerning the determination of the best experimental conditions). (lines 113-118 of the revised version).
- The addition in the Supplementary Methods file of a new paragraph providing the results of the preliminary experiments concerning the incubation time of the selective broth and the reading time of the BLT providing the highest sensitivity, thus retained in the final “BLESSED protocol”. We reproduce here for the reviewers, the two additional figures provided in this online supplementary file, accessible for the readers, summarizing the choices we made for the BLESSED protocol.

5. It may be a little misleading to just point out the 100% sensitivity, when the comparisons in Table 4 and 5 show something different.

We have carefully checked the content of Tables 4 and 5, and we did not see incoherent results. Indeed, Table 3 contains the results of the clinical evaluation of the BLESSED protocol when compared with the culture of the rectal swab medium on ESBL chromID® agar plate (the reference method). These results are summarized in the second column of Table 5 (“Compared with rectal swab culture”). Along these lines, sensitivity was calculated from 22 samples positive with the BLESSED protocol among the 23 samples showing an ESBL-PE in culture of the rectal swab medium = 22/23 = 96%, as indicated in Table 5.

Then, Table 4 contains the results of the clinical evaluation of the BLESSED protocol when compared with the culture of the post-enrichment broth on ESBL chromID® agar plate. Indeed, we observed that more ESBL-PE containing samples were detected when culturing the post-enrichment broth, as a consequence of a lack of sensitivity of the reference method. Thus, this time, sensitivity was calculated from 28 samples positive with the BLESSED protocol among the 31 samples showing an ESBL-PE after culture of the post-enrichment broth = 28/31 = 90%, as indicated in the third column of Table 5.

And so on, with calculations of specificity, and positive and negative predictive values.

Finally, we did not report a “100% sensitivity”, except in the sentence of the result section reporting the results of the *in vitro* performance of the BLESSED protocol: “however, it was positive for all 12 ESBL-PE strains spiked at the 10⁴ CFU/mL inoculum (100% sensitivity)” (lines 159-160 of the original version and 178-179 of the revised version). Further, in the first paragraph of the discussion, we already stated: “Thus, considering *in vitro* and clinical evaluations, the BLESSED protocol had a sensitivity of more than 95% in detecting an ESBL-PE inoculum ≥10⁴ UFC/ml” (lines 213-215 of the original version and 233-235 of the revised version), not pointing out “100% sensitivity”. Consequently, we believe we have not overstated the interpretation of our results.

Reviewer #2

1. In this manuscript, Gallah et al. describe the performance of a rapid diagnostic assay to detect ESBL-producing organisms in the gastrointestinal tract (GI) compared to culture on a chromogenic agar with and without enrichment broth. I hope the authors view these comments as being constructive and designed to help strengthen their manuscript.

We thank the reviewer for this comment and we assure her/him that we have taken her/his remarks as constructive elements.

Major Comments:

2. Introduction: The first section of this manuscript includes a few lines of introducing the readers to the β-LACTA test (lines 62-65), but there is no information on how the test functions. Is it a molecular test, antigenic test, phenotypic, enzymatic, colorimetric? Please provide a brief description of the β-LACTA test.

Following this comment and comment #2 of reviewer #1, we added two sentences briefly explaining how the β-LACTA test functions, as follows: “In contrast, phenotypic tests are simple to use and less expensive. The β-LACTA™ test (BLT) (Bio-Rad Laboratories, Marnes-La-Coquette, France) is a phenotypic test based on the cleavage of a chromogenic cephalosporin, turning the substrate from

yellow to red in the presence of GNB strains producing a beta-lactamase able to hydrolyze 3rd generation cephalosporins. Thus, the test turns positive mainly in the presence of ESBL-PE; however, the test also reacts to some carbapenemase-producing and AmpC-hyperproducing GNB. BLT has shown high diagnostic performance for the rapid identification of ESBL-PE when performed on isolated bacterial colonies (17,18) or directly on bacterial pellets from urine or bronchial aspirate samples (12,14). However, it has never been assessed for the rapid detection of ESBL-PE intestinal colonization.” (lines 69-77 of the revised manuscript).

3. Methods: Part of the technical optimization procedure could be moved to the main body of the manuscript as it gives important insight as to how the test functions. The methods section should include a more detailed description of how the β -LACTA test works.

Following the reviewer’s comment, we fully reworded the first paragraph of the “technical optimization procedure” section. We added some sentences from the Supplementary Methods file, and better explained how the BLT works, as follows: “We first assessed the performance of BLT directly performed on bacterial pellets obtained from the centrifugation of 1 mL of rectal swab medium (*see Supplementary Methods online for additional details*). The pellets were completely homogenized in one drop (approximately 50 μ L) of each reagent of the BLT assay and incubated at 37 °C. This “direct” procedure showed poor diagnostic performance in detecting ESBL-PE (sensitivity 60%, 95% confidence interval (CI) (26%–88%), specificity 58% (42%–72%), positive predictive value 24% (15%–37%), and negative predictive 87% (75%–94%)). These results can be explained by the low sensitivity in detecting ESBL activity diluted in the swab transport medium and by the low specificity due to the false positive result related to other bacteria that can turn the BLT positive, such as some bacteria of the anaerobic flora. We then developed a rapid selective subculture technique to increase both the sensitivity and specificity of the procedure.” (lines 83-93 of the revised version).

We hope that the reviewer will understand that we had to present results (about the diagnostic performance) of this very preliminary phase (which is no longer mentioned in the rest of the manuscript) in a section dedicated to methods.

4. Discussion: Recommend adding discussion for clinical labs on how they can implement this test.

We thank the reviewer for this suggestion. According to this comment we added a few sentences at the end of the discussion with regards to the possible implementation of this protocol in the daily practice of clinical labs, as follows: “Second, our protocol requires several manipulations. Although their number is quite limited, it could be hypothesized that the automation of the procedure would increase its diffusion. However, this must be balanced with the fact that the reference screening strategy is also not fully automated. Indeed, plating of rectal swab medium on selective agar plates, reading the culture results of these plates, and, in case of positivity, sub-culturing of isolated strains to determine their antibiotic susceptibility remain mostly manual. Thus, the BLESSED protocol could probably already be implemented in most laboratories, without a significant increase in technical handling time, but allowing the detection of ESBL-PE intestinal colonization from patient sampling to final results in less than a working day for a laboratory technician.” (lines 290-299 of the revised version).

Minor Comments:

5. Line 43-44: Please describe “current conventional screening strategies” or recommend moving to lines 56-58.

According to this comment, we have detailed what “conventional screening strategies” are in the vast majority of centers, as follows: “Current conventional screening strategies are mainly based on culture on selective agar plates, followed by the identification and confirmation of the antibiotic resistance phenotype of the bacterial strains. Some technical advances, such as automated specimen inoculation and plate streaking (6), plate reading (7), and bacterial identification using matrix-assisted laser desorption/ionization time-of-flight (MALDI-TOF) mass spectrometry, allow for saving some time in the entire procedure. However, in any case, the main limitation of such strategies remains the time required to obtain a definite result, which is approximately 48–72 hours (8).” (lines 44-51 of the revised version).

Reviewer #3

1. In the current study, the authors developed a rapid selective enrichment subculture method prior to the commercial β -LACTA test (Bio-Rad Laboratories) to detect the intestinal colonization of Extended-Spectrum Beta-Lactamase-producing Enterobacteriaceae (ESBL-PE) from rectal swab. However, the authors did not carry out a head-to-head experiment to compare the performance of β -LACTA test alone and enrichment subculture with β -LACTA test.

We thank the reviewer for this comment. We first assessed the performance of the β -LACTA test directly performed on pellets obtained by centrifugation of the rectal swab medium, as this would have been the simplest and most rapid protocol to detect ESBL-PE carriage, in less than one hour after sampling. We mentioned this very preliminary phase at the beginning of the method section (lines 72-75 of the original version, lines 83-91 of the revised version).

As detailed in the “Online Supplementary Methods”, the results were bad with many false negative (due to too small inoculums in the eSwab medium without a first step of enrichment) and many false positive (due notably to the coloration of the test medium directly due to the color of the centrifugation pellet contaminated with feces and to the presence of other bacteria that can turn the BLT positive, such as some bacteria of the anaerobic flora). Full results of this very preliminary experiments performed on 55 independent eSwab medium (10 were positive and 45 negative for ESBL-PE, respectively) were:

	eSwab containing ESBL-PE	eSwab not containing ESBL-PE
β taLACTA positive	6	19
β taLACTA negative	4	26

Thus, sensitivity was 60% 95% CI (26%-88%), specificity 58% (42%-72%); the positive predictive value was 24% (15%-37%) and the negative predictive value 87% (75%-94%). This simple protocol was therefore unusable in clinical practice.

In order to be clearer, we added in brackets the values of the diagnostic performance in the main manuscript. In addition, following several comments from other reviewers, we modified this paragraph, as follows: “We first assessed the performance of BLT directly performed on bacterial pellets obtained from the centrifugation of 1 mL of rectal swab medium (see Supplementary Methods online for additional details). The pellets were completely homogenized in one drop (approximately 50 μ L) of each reagent of the BLT assay and incubated at 37 °C. This “direct” procedure showed poor diagnostic performance in detecting ESBL-PE (sensitivity 60%, 95% confidence interval (CI) (26%–88%), specificity 58% (42%–72%), positive predictive value 24% (15%–37%), and negative predictive 87% (75%–94%)). These results can be explained by the low sensitivity in detecting ESBL activity diluted in the swab transport medium and by the low specificity due to the false positive result related to other

bacteria that can turn the BLT positive, such as some bacteria of the anaerobic flora. We then developed a rapid selective subculture technique to increase both the sensitivity and specificity of the procedure.” (lines 83-93 of the revised version).

2. Although detection of ESBL-PE is a clinically important issue, it is hard to say that this research has new breakthroughs. In my opinion, I think it's just a small modification for the original protocol.

We agree that early detection of ESBL-PE is a clinically important issue. We therefore respectfully disagree with the reviewer, and believe that our protocol is an innovative strategy that allows such detection in less than five hours, with great simplicity of execution; making it feasible in all clinical laboratories. Along these lines, we believe that combining a rapid selective enrichment sub-culture and the implementation of a betaLACTA® test on the resulting bacterial pellets, is not just a “small” modification for the original protocol. Indeed, the validated protocol recommended by the manufacturer is to perform the BLT on freshly isolated bacterial colonies obtained after at least 16h of culture on agar plates.

Major:

3. Native English editing recommended

We thank the reviewer for this suggestion. The revised version of the manuscript has been entirely proofread and corrected by a native English speaker.

4. The term "digestive colonization" could be changed to intestinal colonization or intestinal colonization of ESBL-PE.

According to the reviewer's comment, the term “digestive colonization” has been changed for “intestinal colonization” at each occurrence.

Minor:

5. Add the full name of "BLT test" and "ESBL-PE" in the supplementary methods.

We thank the reviewer for her/his vigilance. Full names were added at their first occurrence and abbreviations introduced before their use, as follows:

- “The β -LACTA® test (BLT) test was performed on bacterial pellets [...]” (first paragraph)
- “Diagnostic performances of this direct procedure were then evaluated by comparison with the direct culture of rectal swab medium on the chromID® extended spectrum beta-lactamase (ESBL) plates. The test was positive for 6 out of the 10 ESBL-producing *Enterobacteriaceae* (ESBL-PE) identified in culture [...]” (following sentence).

December 4, 2022

Dr. Marc Garnier
Assistance Publique - Hopitaux de Paris
Département d'Anesthésie-Réanimation et Médecine Périopératoire Sorbonne Université Rive Droite - Site Tenon
4 rue de la Chine
Paris 75020
France

Re: Spectrum02959-22R1 (Five-hour detection of intestinal colonization with extended-spectrum beta-lactamase-producing Enterobacteriaceae using the β -LACTA[®] phenotypic test: the "BLESSED study")

Dear Dr. Marc Garnier:

Your manuscript has been accepted, and I am forwarding it to the ASM Journals Department for publication. You will be notified when your proofs are ready to be viewed.

Sincerely,

Tulip Jhaveri
Editor, Microbiology Spectrum
